# Fatigue Detection with Spatial-Temporal Fusion Method on Covariance Manifolds of Electroencephalography

**DOI:** 10.3390/e23101298

**Published:** 2021-09-30

**Authors:** Nan Zhao, Dawei Lu, Kechen Hou, Meifei Chen, Xiangyu Wei, Xiaowei Zhang, Bin Hu

**Affiliations:** 1Gansu Provincial Key Laboratory of Wearable Computing, School of Information Science and Engineering, Lanzhou University, Lanzhou 730000, China; zhaon20@lzu.edu.cn (N.Z.); houkch20@lzu.edu.cn (K.H.); chenmf20@lzu.edu.cn (M.C.); xywei2020@lzu.edu.cn (X.W.); 2School of Optics and Electronics, Beijing Institute of Technology, Beijing 100081, China; ludw18@lzu.edu.cn; 3CAS Center for Excellence in Brain Science and Institutes for Biological Sciences, Shanghai Institutes for Biological Sciences, Chinese Academy of Sciences, Shanghai 200031, China

**Keywords:** fatigue detection, electroencephalography, covariance matrices, SPDNet, stein divergence, RNN

## Abstract

With the increasing pressure of current life, fatigue caused by high-pressure work has deeply affected people and even threatened their lives. In particular, fatigue driving has become a leading cause of traffic accidents and deaths. This paper investigates electroencephalography (EEG)-based fatigue detection for driving by mining the latent information through the spatial-temporal changes in the relations between EEG channels. First, EEG data are partitioned into several segments to calculate the covariance matrices of each segment, and then we feed these matrices into a recurrent neural network to obtain high-level temporal information. Second, the covariance matrices of whole signals are leveraged to extract two kinds of spatial features, which will be fused with temporal characteristics to obtain comprehensive spatial-temporal information. Experiments on an open benchmark showed that our method achieved an excellent classification accuracy of 93.834% and performed better than several novel methods. These experimental results indicate that our method enables better reliability and feasibility in the detection of fatigued driving.

## 1. Introduction

The World Health Organization (WHO) stated that subjectively experienced dimensions, such as fatigue and chronic fatigue syndrome (CFS), have been included in the concept of health. Previous works have indicated that chronic fatigue may accompany with a series of physical symptoms [1] such as insomnia, memory loss, and inattention, which also bring out psychiatric diseases like depression and anxiety [2,3,4]. In more serious cases, these physical and mental illnesses may result in a vicious cycle. Besides these negative influences on human health, fatigue also impacts other aspects of human life. For instance, fatigue driving can greatly interfere with a driver’s judgment and response time, thus deeply threaten drivers’ safety. According to statistics mentioned by Li et al. [5], although 57% of drivers believed that fatigued driving is a serious problem, 20% of drivers have fallen asleep or napped while driving in the past year. Maclean et al. referred to some research and arrived at the conclusion that 20–30% of traffic accidents are related to fatigued driving [6]. Thus, effective and reliable fatigue detection, which is an urgent need, can keep drivers away from most traffic accidents. Thanks to the improvement of the computing performance of computers and servers, it is possible to process and analyze data in real-time. Feature engineering and machine learning (ML) algorithms have become a major focus of research on psychological disease detection [7,8], emotion recognition [9], and fatigue detection [10,11,12,13], and related researchers can effectively utilize physiological information such as eye movements, body movements, facial expressions, and signals from modalities such as EEG, electromyography (EMG), electrocardiography (ECG), and galvanic skin reaction (GSR) to monitor the brain activity [14,15].

Recently, the classification of EEG signals by using Riemannian geometry has been proposed by many researchers [16,17,18], and it has become a hot topic to utilize covariance matrices as features of signals. However, most researchers only calculated covariance matrices of the whole trial as the total features, which neglected the information of temporal relation in symmetric positive-definite (SPD) space throughout the duration of a trial. Alternatively, by utilizing the superiority in Riemannian geometry, except for the calculation of distance and mapping, which is the shallow use of Riemannian network, we can extract more effective higher-level features in SPD space [19]. Therefore, our study was centered on the changes in the spatial-temporal relations of signals recorded by different EEG channels, and we investigated whether it is discriminative in the fatigue driving detection. For ease of description of our model, we have defined some concepts. First, all the covariance matrices calculated from EEG signals we used were defined as relation domain, which means they can reflect the relationship between different EEG channels. Then, the relation domain was divided into two subdomains: temporal relation (TR) domain and spatial relation (SR) domain. In the spatial relation domain, the covariance matrix of each trial was calculated, then we took the characteristics of its distribution as the SR-domain features in SPD space. In the temporal relation domain, EEG signals are partitioned into several segments by leveraging the slide window technique, then covariance matrices corresponding to each segment are obtained and flattened. It has been proven that the long short-term memory (LSTM) network could solve the gradient vanishing problem of the basic RNN model [20], and the LSTM network performs better while processing long series problem such as language modeling [21]. Meanwhile, some modified model based on LSTM has been utilized in more scenarios such as real-time traffic flow prediction [22] and 3D action recognition [23,24]. Therefore, we use an LSTM network to obtain the TR-domain features of higher level by modeling the relationships between these flattened matrices. Then, SR-domain features and TR-domain features are fused by the fully connected (FC) layer, and the fused features were fed into the softmax layer to make the final prediction. The experimental results compared with other methods show that our method achieves an accuracy of 93.834% for detecting fatigue state and demonstrate that this method outperforms several advanced methods. The main contributions of our study are listed as follows:It discusses the relationship between different EEG channels and explores the changes of temporal dynamics of this information, which further contributes to the extension of signal processing and modeling conception.It combines two kinds of spatial features on the SPD space to improve the classification performance of different driving states.It proposes a state-of-the-art fusion method, which utilizes the spatial-temporal information of covariance matrices on the Riemannian manifold, and achieves the mission of detecting fatigue driving state effectively.

The remainder of this paper is organized as follows. In Section 2, the construction of our method and extraction of TR and SR features are reported. In Section 3, we present the description of dataset and report the experimental results, we also analyze the relationship between brain state and the covariance matrix. Finally, in Section 4, we conclude our work and discuss the possibility of future expansion and application.

## 2. Methodology

As we defined before, in the relation domain constructed by the covariance matrices, we can learn the relationships between signals of different EEG channels by processing these matrices. The covariance matrices obtained from EEG signals have been shown to be widely used in many areas [25,26], which also prove that the relation domain we defined could greatly contribute to the feature extraction. As proven in [27], the driver’s mental state while driving can be revealed by the information of some specific regions of the brain under the cortices, and the design of our method to obtain some specific features in the relation domain can be supported by understanding these sources of information. The covariance matrices in SR domain reflect the characteristics of the whole trial, which means the EEG signals of different states are analyzed from a macro perspective. According to some advances in technology of brain–computer interface [26,28], the feasibility of classification method based on the spatial covariance matrix can be demonstrated. However, just as we mentioned before, most studies only considered the spatial features captured from the whole trial and neglected the temporal information over time. Moreover, in practical applications, the spatial distribution of covariance matrix is more concerned, while the temporal information of covariance characteristics of EEG signal has not been deeply studied. In time-series tasks such as action recognition, forecasting, and virtual driving task, the temporal changes over time contain more information that can be well used. We assumed that the covariance matrices from the partitioned signals can capture the relationship between brain regions over time, and based on this, the temporal domain features of covariance of EEG signal are extracted and investigated in our study. Thus, as shown in Figure 1, we propose the framework contains spatial features and temporal features and fuses them together, which will be introduced in detail in the remainder of this section.

### 2.1. Features of Spatial Relation Domain

Huang et al. [19] proposed a novel and special Riemannian network architecture in 2016. This network could project the SPD matrices to a more smoother SPD space because of the depth nonlinear characteristic mapping. Through the end-to-end method, the model can automatically learn the optimal mapping mode, so as to extract the higher-level information of the covariance matrix, which greatly improves the classification accuracy of the model. Figure 2 shows the pipeline of the SPD network; the first layer of this network is called Bilinear Map Layer, which is utilized to generate covariance matrices in the SPD space with more discrimination. However, it is possible that some large eigenvalues of the original matrices may be reduced to very small number after being processed by Bilinear Map Layer. These very small eigenvalues may affect the positive-definite properties of matrices, which violates the principle of SPD space. Thus, the Eigenvalue Regularization Layer is set to process this issue by replacing these special eigenvalues with the threshold. In order to apply the classical Euclidean computations, the Logarithmic Eigenvalue Layer utilizes the logarithm operation of matrices proposed by Arsigny [29] to reduce the matrices to a flat space. In this study, we applied two Bilinear Map Layer, one Eigenvalue Regularization Layer, and one Logarithmic Eigenvalue Layer to construct the SPD network, which uses the whole-trial covariance matrices as input.

Although the SPD Network extracts the higher-level information of the covariance matrix, when we reshape it into a 1-D vector, it may lose some information of the original matrix. It has been verified by Congedo [18] that the sample distribution of each class could be reflected by the geometric mean of that class with proper matrices based on SPD space. In addition, the classification algorithm could achieve reliable results by utilizing the distance information between the mean of classes and each sample, which also keeps the distance information of matrices in the original SPD space. Therefore, in our study, we believe that by using *symmetric Stein divergence* [30], the vigilant level of the driver can be distinguished to some extent. The Stein divergence is used to measure the distance between the centers of two classes (drowsy and vigilant) and each sample, which could supplement the information of the original matrix by utilizing the distance information.

The whole-trial matrices of SR domain are assumed to be zero-centered, which is defined as *C*:(1)$$C=\frac{1}{Q-1}XX^{T},X\in R^{N\times Q}$$
where *X* denotes the EEG signal of one whole trial, Q=9 s × 250 Hz =2250 denotes the sampling points of trial, and N=30 indicates the number of EEG channels As proven by Sra [30], the center of these covariance matrices should be subject to the condition as follows:(2)$$\hat{Center}=\underset{Center}{\arg\min}\left(\sum_{i=1}^{M}\delta_{R}^{2}\left(Center,C_{i}\right)\right)$$
where *M* denotes the number of covariance matrices, and in the training set, the covariance matrix of one single trial is calculated and defined as $C_{i}$, $\hat{Center}$ denotes the center of *M* trialwise covariance matrices. Moreover, the first order of (Equation 2) should be satisfied with
(3)$$▿\left(Center\right)=\sum_{i=1}^{M}{\left[\frac{Center+C_{i}}{2}\right]}^{-1}-MCenter^{-1}=0$$

According to the work in [31], the (Equation 3) has no analytical solution, thus as illustrated in Algorithm 1, we utilize the iterative approximation to solve the problem.
**Algorithm 1** Centre of Covariance Matrices**Input:** $C_{i},i\in 1,M$**Output:** The Centre of $C_{i},i\in 1,M$1:initial: *Centre* = $\frac{1}{M}\sum_{i=1}^{M}C_{i}$, *iterator* = 0; thre = $1\times 10^{-9}$; Maxiterator = 50;2:**repeat**3:    $C_{new}={\left[\frac{1}{M}\sum_{i=1}^{M}{\left(\frac{C_{i}+Centre}{2}\right)}^{-1}\right]}^{-1}$;4:    $cri={∥C_{new}-Centre∥}_{F}$;5:    $Centre=C_{new}$;6:    iterator=iterator+1;7:**until***cri* <*thre*
**or**
*iterator*> *Maxiterator*

In Algorithm 1, the Frobenius norm cri is defined to represent the difference between $C_{new}$ and Center. We define the Maxiterator as the maximum number of iterations and thre as the threshold to satisfy the requirement of the convergence stability. Similar to the Eigenvalue Regularization Layer of SPD network, to ensure the positive-definite properties of the covariance matrices, each covariance matrix *C* is regularized as
(4)$$C=U_{c}max\left(\Lambda_{c},\epsilon I\right)U_{c}^{T}$$
where epsilon is the threshold we set ($1\times 10^{-8}$), and *I* represents the identity matrix. The eigenvalues $\Lambda_{c}$ and eigenvectors $U_{c}$ are calculated by applying eigendecomposition:(5)$$C=U_{c}\Lambda_{C}U_{c}^{T}$$

The variate $max(\Lambda_{c},\epsilon I)$ of (Equation 4) is a diagonal matrix *E* whose diagonal elements are defined as
(6)$$E\left(i,i\right)=\left\{\begin{array}{c} \Lambda (i,i),\Lambda (i,i)>\epsilon \\ \epsilon ,\Lambda (i,i)\leq \epsilon \end{array}\right.$$

In both training set and testing set, the distances between the covariance matrix $C_{j}$ of each trial and the centers of the two classes trials with vigilant or drowsy labels are calculated as
(7)$$\delta_{R}\left(C_{j},C_{V/D}\right)=\sqrt{\log\left(\frac{\left|(C_{j}+C_{V/D})/2\right|}{\sqrt{|C_{j}C_{V/D}|}}\right)}$$

Finally, according to the work in [16], we took these distances information as another kind of feature in the SR domain. Then, we fuse the features obtained from SPD network and Stein divergence as the total features of SR domain.

### 2.2. Features of Temporal Relation Domain

In this study, we assumed that the variations in the relations between different regions of the brain over time can be captured by the partitioned EEG signals and their consecutive covariance matrix. The driving fatigue state can be reflected by learning these variations. Figure 3a shows the sample of signals from each trial. The 9 s signals were divided into seven segments with a time window of 3 s and a step of 1 s. Thus, we define the covariance matrix corresponding to the *t*-th segmented signal as follows:(8)$$C_{t}=\frac{1}{d-1}X_{t}X_{t}^{T},t\in 1,2,\cdots ,7$$
where *N*
=30 means the channel number, *d* = 3 s × 250 Hz = 750 indicates the sampling points. Thus, the time series of the *t*-th segment with *N* channels are represented by $X_{t}$, which is assumed to have a mean of zero.

As shown in Figure 3b, according to the symmetry of the covariance matrix, we take the diagonal elements and the lower triangular elements to reconstruct the matrix into a one-dimension vector $m_{t}$, which has 465 elements. Sabbagh [32] has proved that such a flattened covariance matrix can simulate the linear features of the brain source, thus avoiding the problem of solving the inverse matrix directly [32]. Let *O* denote the number of linearly independent brain sources, thus the transformation matrix $A\in R^{N\times O}$ is required to connect signals of EEG channels $X_{t}$ and brain sources $E_{t}\in R^{O\times d}$ by using the equation $X_{t}=AE_{t}$. That means we can use $E_{t}$ and *A* to represent the covariance matrix $C_{t}$ as $C_{t}=\frac{1}{d-1}AE_{t}Et^{T}A^{T}$. Sabbagh et al. [33] suggested in their study that the signals of the brain sources are linearly independent with each other, which means the covariance of two different sources is zero. Thus, the matrix $E_{t}Et^{T}$ is a diagonal matrix, whose diagonal elements are the variances. By utilizing this method, without calculating the transformation matrix explicitly, we can extract the power of brain sources linearly. Accordingly, we defined the features of the 9 s trial in TR domain as follows:(9)$$\left[m_{1},m_{2},\cdots ,m_{7}\right]$$
the power of brain sources can be represented by $m_{t}$, and the dynamic changes over time could be learned from the series above.

By taking the advantage of RNN, which is always used to process the time series data by exploring the comprehensive underlying information over time. Thus, we choose LSTM networks to extract TR-domain features at a high level. As shown in Figure 4, in our model, each RNN layer is constructed with 7 LSTM units. Several previous works [34,35] have achieved excellent performance, and as proven in paper [36], such a well-designed RNN structure could obtain the underlying information of the input sequences such as long-term dependencies. In the TR domain, a 2-layer RNN model is designed to capture the intrinsic information of covariance matrices over time In Figure 4, the first RNN layer process the series of (Equation 9) and output the hidden variable sequence as $\left[h_{1},h_{2},\cdots ,h_{7}\right]$, where $h_{t}$ and $h_{t+1}$ represent the consecutive hidden state. The hidden variable sequence is also the input of the second RNN layer, which denoted by $\left[h_{1}^{{}^{\prime }},h_{2}^{{}^{\prime }},\cdots ,h_{7}^{{}^{\prime }}\right]$. Thus, we got $h_{7}^{{}^{\prime }}$, which is the final output of the RNN, and we fused it with the features from the SR domain.

### 2.3. Fusion and Optimization

Considering that the features from both TR domain and SR domain contribute to the final fatigue detection, as illustrated in Figure 1, we fused the $\delta_{R}\left(C_{j},C_{V}\right)$ and $\delta_{R}\left(C_{j},C_{D}\right)$, which calculated based on Stein divergence, features obtained from SPDNet and features of TR domain captured by the RNN together. Referring to [37], the spatial-temporal joint optimization method is illustrated in Algorithm 2. We optimize the parameters alternately in each iteration. Specifically, when optimizing the parameters of SPD network, we fix the parameters of RNN, and vice versa. Different from the other parameters, the bias and weight of fully connected layer are updated in the end of each iteration. As shown in the algorithm, $\left[X_{L}^{t_{l}},X_{S}^{t_{s}},X_{D}\right]$ denotes the vector of fused features from both domains, where $X_{L}$ is the feature of the TR domain, $X_{S}$ is the vector transformed from the trial-wise SPD covariance matrix after being mapped by the SPD network, and $X_{D}$ is the feature vector $\left[\delta_{R}\left(C_{j},C_{V}\right),\delta_{R}\left(C_{j},C_{D}\right)\right]$ introduced previously. The mapping operation by FC layer is represented by $F\left(\left[X_{L}^{t_{l}},X_{S}^{t_{s}},X_{D}\right],\omega_{F}^{t}\right)$, where $\omega_{F}^{t}$ is the weight parameters of FC layer. Besides, we utilize the cross-entropy as our loss function $£\left(Y,f_{t}\right)$. In order to record the number of iterations of optimization, $t_{s}$ and $t_{l}$ are defined in the optimization of SPD network and RNN, separately, while *t* indicates the total number of iteration.

In summary, we obtain the SR-domain features, which are the fused vectors consist of the distances information between the covariance matrices of a single trial and the center of class based on Stein divergence as well as the reshaped vector obtained from the SPD network. Meanwhile, the TR-domain features are the final output of the 2-layer RNN by processing several time series segments of a single trial. Then, we concatenate the features and get the ultimate predictions by feeding the features into the fully connected layer and softmax layer.
**Algorithm 2** Spatial-temporal joint optimization algorithm**Input:** $X_{L}$: input of LSTM; $X_{S}$: input of SPDNet; $X_{D}$: input of Stein divergence; *Y*: Training set label**Output:** $\omega_{L}$: parameters of RNN; $\omega_{S}$: paremeters of SPDNet; $\omega_{F}$: parameters of FC layer1:initial Maxite = 50; $t_{l}=1$; $t_{s}=1$;2:**for** each t∈[1,Maxite]
**do**3:    **if**
t%2=1
**then**4:        $X_{L}^{t_{l}+1}=L\left(X_{L}^{t_{l}},\omega_{L}^{t_{l}}\right)$5:        $f_{t}=F\left(\left[X_{L}^{t_{l}+1},X_{S}^{t_{s}},X_{D}\right],\omega_{F}^{t}\right)$6:        $Loss=£\left(Y,f_{t}\right)$7:        $▿\omega_{L}^{t+1}=\frac{\partial Loss}{\partial f_{t}}\cdot \frac{\partial f_{t}}{\partial X_{L}^{t_{l}+1}}\cdot \frac{\partial X_{L}^{t_{l}+1}}{\partial \omega_{L}}$8:        ${\tilde{\omega }}_{L}^{t_{l}+1}=\omega_{L}^{t}-\eta_{L}▿\omega_{L}^{t_{l}+1}$9:        $t_{l}\leftarrow t_{l}+1$10:    **end if**11:    **if**
t%2=0
**then**12:        $X_{S}^{t_{s}+1}=S\left(X_{S}^{t_{s}},\omega_{S}^{t_{s}}\right)$13:        $f_{t}=F\left(\left[X_{L}^{t_{l}},X_{S}^{t_{s}+1},X_{D}\right],\omega_{F}^{t}\right)$14:        $Loss=£\left(Y,f_{t}\right)$15:        $▿\omega_{S}^{t+1}=\frac{\partial Loss}{\partial f_{t}}\cdot \frac{\partial f_{t}}{\partial X_{S}^{t_{s}+1}}\cdot \frac{\partial X_{S}^{t_{s}+1}}{\partial \omega_{S}}$16:        ${\tilde{\omega }}_{S}^{t_{s}+1}=\omega_{S}^{t}-\eta_{S}▿\omega_{S}^{t_{s}+1}$17:        $t_{s}\leftarrow t_{s}+1$18:    **end if**19:    $▿\omega_{F}^{t+1}=\frac{\partial Loss}{\partial f_{t}}\cdot \frac{f_{t}}{\partial \omega_{F}}$20:    ${\tilde{\omega }}_{F}^{t+1}=\omega_{F}^{t}-\eta_{F}▿\omega_{F}^{t}$21:**end for**

## 3. Experimental Results and Discussion

### 3.1. Dataset and Data Processing

#### 3.1.1. Experimental Paradigm

In this study, we use the open benchmark dataset, which consists of EEG signals collected during the sustained-attention driving task [38] based on the experimental paradigm of immersive driving proposed by Huang et al. [39]. This paradigm and the dataset were also used in related works [40,41,42,43,44], and the validity and reliability have been verified by these works with effective result. As mentioned in [38], the study of the dataset we used was performed in strict accordance with the recommendations in the Guide for the Committee of Laboratory Care and Use of the National Chiao Tung University, Taiwan. Besides, the study mentioned was also approved by The Institutional Review Board of the Veterans General Hospital, Taipei, Taiwan. As shown in Figure 5, the participants were required to drive at a constant speed of 100 km/h in a virtual night-time scenario, and the car will randomly drift out of the lane at the speed of 5 km/h. Subjects were asked to steer the car back to the center of the lane as quickly as possible. Then, the time spent driving back to the original lane, which defined the reaction time, was used to represent the drowsy level of the subject. As shown in Figure 6, we can see a clear difference between the two states, and most reaction times of subjects in the vigilant state were around the average, which is also significantly smaller than that of the fatigue state. Thus, we believe that most people are more likely to behave differently between the vigilant and fatigue state.

#### 3.1.2. Definition of Labels

In our study, in order to include the information which could reflect the driving state, we capture the data from 1 s before the response offset to 8 s after that as the data of one trial, which contains the deviation onset and response onset. We defined the reaction time mentioned before as local reaction time (RT). As defined in [45], we average several local RT values around the current trial and define the average as the global RT, by which the level of drowsy state while driving could be represented. Thus, the global RT for the *l*-th trial was defined as
(10)$$globalRT\left(l\right)=\sum_{i=l-n}^{l+n}\frac{localRT(i)}{2n+1}$$
where *n* was set to 2 in our study. As shown in Figure 7, we defined the following vigilant and drowsy labels, where $RT_{l}$ denotes the local reaction time and $RT_{g}$ denotes the global reaction time: (11)$$Trial\left(i\right)\in \left\{\begin{array}{c} \left\{RT_{l}\left(i\right)\leq 0.62\right\}\cap \left\{RT_{g}\left(i\right)\leq 0.62\right\},vigilant\\ \left\{RT_{l}\left(i\right)\geq 1.5\right\}\cap \left\{RT_{g}\left(i\right)\geq 1.5\right\},\;\;drowsy \end{array}\right.$$

Accordingly, with a total number of 1433 trials, 739 trials were defined as the vigilant labels and 694 trials were defined as the drowsy labels. The length of each trial was 9 s, and the number of trials of each subject varies from 2 to 179.

#### 3.1.3. Data Preprocessing

According to the work in [38], 27 volunteers were recruited to participate in a 90-min task of sustained virtual attention driving while 62 EEG recordings were collected. The original signals were collected using a system with 32 electrodes, which includes 2 reference electrodes, and digitised at 500 Hz with 16-bit resolution. For the purpose of eliminating the noise, we downsampled these EEG signals to 250 Hz and used bandpass filtering to retain the information between 1 and 50 Hz. In order to remove the artifacts that we are not interested in, such as muscle activity, eye movement, blinking, and ECG signals, we utilize Independent Component Analysis (ICA) to extract the pure EEG signals.

### 3.2. TR-Domain Experiments

#### 3.2.1. Validity Analysis of TR-Domain Features

To visually investigate the temporal dynamics of the relationship between EEG channels, the reshaped flatten covariance matrices of one single trial were combined into a 7 × 465 matrix defined as *M*, whose column indicates the dynamic changes of one pair of EEG channels. By averaging the matrices *M* of all trials belonging to different classes, we get $M_{V}$ and $M_{D}$ to represent the dynamic changes of 739 trials with vigilant labels and 694 trials with drowsy labels. The Pearson’s correlation coefficient (PCC) was used to calculate the correlations between different columns of *M*, and the PCC could reflect whether we can learn useful identifying information from the variations. Considering the tradeoff of computational complexity, we chose only 2, 3, 5, and 7 channel pairs and all pairs and focused on comparisons of them, which can be used to prove the extensibility and practicability of our model. Given the chosen number of EEG channel pairs, we defined the measurement ζ for each possible combination of different columns of *M* as below to check the inter-class and intra-class correlations:(12)$$\zeta =\frac{\Gamma +\Theta }{\Phi +\Upsilon }$$
where Γ, Θ, Φ, and Υ are defined as
(13)$$\Gamma =\sum_{i,j\in \chi_{f}^{V},i\neq j}\exp\left(\phi \left(p_{i}^{v},p_{j}^{v}\right)-\omega \right)$$
(14)$$\Theta =\sum_{i,j\in \chi_{f}^{D},i\neq j}\exp\left(\phi \left(p_{i}^{d},p_{j}^{d}\right)-\omega \right)$$
(15)$$\Phi =\sum_{i\in \chi_{f}^{V},j\in \chi_{f}^{D}}\exp\phi \left(p_{i}^{v},p_{j}^{d}\right)$$
(16)$$\Upsilon =\sum_{i,j\in \chi_{f}^{V},k,l\in \chi_{f}^{D}}||\phi \left(p_{i}^{v},p_{j}^{v}\right)-\phi \left(p_{k}^{d},p_{l}^{d}\right){\left|\right|}^{2}$$

Let $\chi^{V}$ and $\chi^{D}$ denote all possible combinations of a given number of column pairs in $M_{V}$ and $M_{D}$, respectively. We define the combination function *f* to represent the set of combinations of columns, and the $\chi_{f}^{V}$ and $\chi_{f}^{D}$ indicate the same elements in $\chi^{V}$ and $\chi^{D}$, respectively. The *i*-th column in $M_{V}$ and $M_{D}$ were presented by $p_{i}^{v}$ and $p_{i}^{d}$, which belong to the column set in $\chi_{f}^{V}$ and $\chi_{f}^{D}$. The PCC between $p_{i}^{v}$ and $p_{i}^{d}$ is denoted by $\phi (p_{i}^{v},p_{i}^{d})$. For example, we want to choose 3 columns to calculate the measurement, so we can randomly choose 3 columns in $M_{V}$ and $M_{D}$. Then, the $\chi^{V}$ and $\chi^{D}$ consist of ${}_{465}C_{3}$ combinations, respectively. For a given combination fuction *f*, there are 3 corresponding columns in $\chi_{f}^{V}$ or $\chi_{f}^{D}$, respectively. Here, ω is set to be 0.9 in our work, which is the threshold representing the strong correlation criterion between two columns. As shown in (Equation 13) and (Equation 14), Γ and Θ represent the intra-class correlations of the same combination function *f* in $\chi^{V}$ and $\chi^{D}$. Φ denotes the inter-class correlation of the same combination *f* in $\chi^{V}$ and $\chi^{D}$, and Υ indicates the similarity between the intra-class correlations for the same column pair in $\chi_{f}^{V}$ and $\chi_{f}^{D}$, thus *i* and *k* are the same column, and *j* and *l* are the same column in (Equation 16).

The purpose of this experiment is to ascertain whether we can learn some useful discriminative information from the TR domain; ζ is defined to measure whether the combination we selected meet our requirement. Thus, we aim to find the combinations with relatively high value of ζ, which are achieved in cases of high values of Γ and Θ and low values of Φ and Υ, which means the combination with high value of ζ has relatively weak inter-class correlations and strong intra-class correlations. These combinations with weak inter-class correlations show the difference between vigilant and drowsy driving state. To intuitively illustrate the results based on the measurement ζ, Figure 8 clearly shows the dynamic variations in the relationship of different channels over time in the two states. The trajectories showed in the figure clearly illustrate the truth that the changes of the same class are similar to each other but the variations of different classes are completely different. Furthermore, Table 1 presents quantitative information on the intra-class and inter-class correlations. As shown in Table 1, the intra-class correlations are generally higher than the inter-class correlations.

The results of this section are consistent with the hypothesis of the previous section Features of Temporal Domain, which indicates that the variations of the relationship between different channels over time play an important role in the classification. Based on research on this hypothesis, the feasibility as well as reliability of developing such a method that utilizes TR-domain features can be further proven.

#### 3.2.2. Comparison of Different Number of Channel Pairs

To evaluate the effectiveness of our method and avoid data leakage, we took the advantage of the leave-one-subject-out strategy to conduct the experiments on the open dataset mentioned in Dataset and Data Processing section, which also proves that our model performs well on subject independence. Considering the advantage of TensorFlow framework, we implemented our model in it, and the rectified linear unit (ReLU) and cross-entropy loss was set as the activation function and loss, respectively. To avoid overfitting, the early stop trick was applied in our experiments. Besides, we use the evaluation index as follows to evaluate our fatigue detection model: accuracy, sensitivity, specificity, and F1 score.

As shown in Figure 9, we selected different numbers of columns to conduct the classification experiment. The results show that the classification accuracy improved with the increase of the selected numbers, which indicates that model with more EEG channel pairs could obtain more comprehensive information. Therefore, classification using all EEG channel pairs outperforms that using the subset of channel pairs. By using the paired samples *t*-test, we found that there are significant differences between the different number of channel pairs because the *p*-value is smaller than 0.01. As shown in Figure 9, the model selected all channel pairs achieved the highest accuracy. With the decrease of channel number, the classification accuracy also decreased by about 4–10%. Considering the volume conduction effects in EEG [46], the signal of each EEG channel is not generated by a single neuron but by the discharge of many neurons. Therefore, with the increasing of the selected number of channel pairs, the linear power features of the neurons under test will increase, which can be reflected by the classification accuracy.

### 3.3. Comparative Experiments

To prove the effectiveness of our model by fusing features from the spatial and temporal domain, we compared several novel methods with ours. These methods also calculated the covariance matrices and utilized them as features to construct their models. Consistent with our method, the same leave-one-subject-out strategy and experimental settings were used in the other selected methods, in which accuracy, specificity, sensitivity, and F1 scores were utilized for evaluation.

By using the novel neural network, Mehdi Hajinoroozi et al. [47] proposed a method based on convolutional neural network (CNN) which utilized covariance matrices as the input features. In our experiments, we named that method as COV_CNN to compare with ours. Similarly, we abbreviated the method using CNN and RNN parallelly as CNN_RNN [48], which also used covariance matrices as input. Referring to the work in [45], we also choose a common method without using the covariance matrix. The FRE method cited in our manuscript utilized power spectral density (PSD) matrix and CNN to construct the model. Considering the information contained in the TR domain, we called the method with all channel pairs in the previous experiment as the TR domain classifier (TRDC) and designed an experiment based on that. Finally, in order to investigate the effectiveness of using only Stein divergence or SPDNet features to fused with TR-domain features, we designed a classifier using Stein divergence and TR-domain features (SDTR) and that using SPDNet and TR-domain features (SNTR), and then conducted classification experiments utilizing them, respectively. The constructions of these methods are as follows.

1.COV_CNN: Consistent with the work in [47], a CNN was set to extract the information of whole-trial covariance matrices. The kernel size of CNN with 1 pooling layer were set to be 3×3×10 and 3×3×5, respectively, while the stride was set to be 1.2.CNN_RNN: Referring to the work in [48] and being consistent with our work, a CNN with 2 layers and an RNN with 2 layers are combined parallelly to reconstruct the model. Then, the output features of CNN and RNN are fused together and sent into the softmax layer for final prediction. The kernel sizes of the first and second CNN layer were 3×3×32 and 3×3×64, and the kernel stride was 1.3.FRE: We selected the signal of each channel by bandpass filtering with 1–30 Hz and divided it into 30 bands on average. Then, we utilized short-term fast Fourier transform (STFT) to calculate the PSD features and obtain the amplitude in each band. By processing the signals, each single trial could be denoted by a PSD matrix, whose size is 30×30. Considering that CNN has excellent capabilities of image processing, the obtained PSD matrix was fed into a 3-layer CNN, whose parameters were set as follows: the kernel sizes of each layer were set to be 3×3×1, 3×3×32, and 3×3×64, respectively, and the kernel stride was 1.4.TRDC: The output of RNN is thought to be the final feature of the TRDC model. Without being fused with other features, the output is fed into the fully connected layer. Besides, the parameter settings of this method are the same as the spatial-temporal fusion method we proposed.5.SDTR: Consistent with the method in [31], without fusing with features obtained from SPDNet, we only fused distance features calculated based on Stein divergence and features of the TR domain. Then, we fed this feature vector into the FC layer and softmax layer for prediction. The framework of TR domain is the same as our method.6.SNTR: Following the parameters setting of [37], the features obtained by mapping a single covariance matrix using SPDNet and features of the TR domain were fused together to be fed into the FC layer and softmax layer. Then, we used the prediction for the final classification.

As shown in Table 2, in comparison with our fusion method, the SDTR achieved an average accuracy of 89.280%, and the SNTR achieved an average accuracy of 91.042%, whose performances were reduced by approximately 4.554% and 2.792% compared to that of our method. Thus, it was proven that the SR domain features combined Stein divergence and SPDNet utilizing EEG signals could better reflect the difference between vigilant and drowsy states, which is consistent with the assumption of the previous section SR-Domain Features. TRDC achieved an average accuracy of 86.238% even without considering the features of the SR domain, which indicates that the features of the TR domain contain more discriminative information for the classification of fatigue. This result also shows us the importance of information over time in long-term time series tasks, which is one of the reasons why we select RNN to process this modeling task. However, as shown by the experimental results without asterisks of SDTR and SNTR in Table 2, through the significance test of the evaluation criteria, it was found that there was no statistically significant difference in these evaluation results between our methods and that of the two methods above. We analyzed the possible causes and checked the *p*-value between TRDC and SDTR, TRDC and SNTR, SDTR and SNTR, respectively. The results showed that there was no significant difference between SDTR and SNTR, while the other combinations had significant differences. According to the results of TRDC, we know that temporal information contributes more to the final results than spatial information; thus, we believe that the difference between different feature extraction methods in the spatial domain is not significant. However, when the spatial features were fused with temporal features, the model truly improved the classification accuracy, which can be proven by the experimental results. Furthermore, the spatial features based on the fusion of SPD network and Stein divergence also outperformed TRDC. However, compared with SDTR and SNTR, although it performed better than these two methods, because of the main contributions of TR-domain features, the differences between our method and SDTR as well as SNTR were not significant, which also can be proven by the significance test result between SDTR and SNTR. Therefore, how to improve the significance of the difference is worth to be explored in future work. The mean accuracy achieved by FRE is 75.155%, which is also inferior to that of our method, thus the truth that the relation-domain features contribute more to the classification model than features as PSD matrices can be proven. Compared with the methods that neglected the temporal information in the relation domain such as COV_CNN (the mean accuracy is 81.877%), our methods performed better, whose accuracy improved about 11.957%. Although CNN_RNN also uses an LSTM network to capture the features of covariance matrices, it only captures spatial features due to the use of the whole-trial matrix. As mentioned before, temporal features improves the performance of classification more, therefore the average accuracy of CNN_RNN is only 79.103%, which is reduced by about 14.731% compared to that of our method. In comparison, our method reduces the complexity of the model, avoids the introduction of noise, and improves classification performance by fusing both spatial and temporal features of the relation domain. In other words, learning from all experiments conducted previously, the spatial-temporal fusion method outperforms other methods that only consider spatial information alone.

As the experimental results showed in the section Comparison of Different Number of Channel Pairs, considering the trade-off of computational complexity, time cost, and classification accuracy, we compared the results with the different number of channel pairs and got the conclusion that the model can achieve relatively high accuracy although reducing the number of channel pairs. Thus, the possibility of future online use of our model can be supported by this result. GM Duma et al. [49] also discussed the possibility of whether such an EEG-based detection model could be used in practice to protect the safety of drivers. Some related works [50,51] also mentioned the online use of EEG-based models in the area of driving assistant systems, and the authors of [50] have already tested their system while driving the real car. With high classification accuracy and low time cost, we believed that our model could be used to detect the fatigue state in practice. In future studies, we will test the performance of our algorithm on other datasets.

### 3.4. Brain State Analysis Based on Covariance Matrix

Recent evidence [45] suggests that the locations of the EEG channels represent the distribution of different brain regions and that the covariance matrices capture the relations between different channels. As mentioned in section TR-Domain Features, the covariance matrix can simulate the linear features of the brain source. EEG signals captured from the scalp are superposed signals generated by the discharge of many neurons, then we let *K* denote the number of linearly independent brain sources. Therefore, the set of all neurons is represented as $s\left(n\right)=\left[s_{1}\left(n\right),s_{2}\left(n\right),\cdots ,s_{K}\left(n\right)\right]\in R^{N\times K}$, where *n* is the time-series length of signals produced by the neurons. Let *M* denote the number of EEG channels, $x\left(n\right)=[x_{1}\left(n\right),x_{2}\left(n\right),\cdots ,x_{M}\left(n\right)]$ represents the signals of all channels. As illustrated in [52], in the study of brain sources, we often divide the problems into the forward problems and the inverse problems. The inverse problem is locating the position of the source of evoked potentials from measurements on the surface of the scalp. Thus, the matrix W is required for modeling the brain sources as $s\left(n\right)=W^{T}x\left(n\right)$, which is based on that the EEG signals could be represented by the linear combination of discharge produced by the neurons. In contrast, the forward problem is solved by starting from a given electrical source and calculating the potentials at the electrodes, which means given the current signals s(n), there is a matrix A that could denote the signals of channels as x(n)=As(n)+ε(n), where ε(n) denotes the noise during the conduction. In general, we assume that the noise is linearly independent with each other, thus the covariance $\Sigma_{\varepsilon }=0$. Haufe et al. proposed that ${\mathrm{AW}}^{T}=I$, where I is the identity matrix [53]. However, the conduction matrix is not guaranteed to be a square matrix, so in the process of solving the forward problem, the following transformation should be made:(17)$$\Sigma_{x(n)}=\mathrm{AE}\left[s\left(n\right)s^{T}\left(n\right)\right]A^{T}+\Sigma_{\varepsilon }$$
because $\Sigma_{\varepsilon }=0$, so
(18)$$\Sigma_{x(n)}=\mathrm{AE}\left[s\left(n\right)s^{T}\left(n\right)\right]A^{T}$$
we multiply W on the both sides of Equation (Equation 17):(19)$$\Sigma_{x(n)}W=A\Sigma_{s(n)}$$

As shown in Equation (Equation 19), the covariance characteristics of EEG signals are closely related to the activity of the neurons, which is also the solution of the forward problem:(20)$$A=\Sigma_{x}W\Sigma_{s}^{-1}$$

Sabbagh et al. [33] proposed that the electrical activity of the brain neurons is linearly independent, that is, $\Sigma_{s(n)}=0$. Therefore, it is proven that the flattened covariance matrix can simulate the linear features of the brain source, which could avoid solving the inverse problem directly [32].

According to Equation (Equation 19), when the solution of the inverse problem is determined in the localization of the brain source, the covariance matrix corresponds to the weight matrix of the forward problem, which is closely related to the activity state of the brain. Based on this, we conducted research on the influence of the covariance matrix of EEG signals on the activity of the brain neurons under fatigue and vigilant conditions.

As shown in Figure 10a, we assumed that the number of neurons in brain is 364; $\Sigma_{x}\in R^{30\times 30}$ denotes the covariance matrix obtained from the EEG signals. Under the condition that the head model is fixed, we utilized BrainStorm to obtain the solution of the inverse problem of brain source localization, which is represented by the transform matrix $W\in R^{30\times 364}$. Thus, we obtained the electrical activity of the neurons $\hat{S}\in R^{364\times 2250}$ corresponded with each EEG segment, and the covariance matrix of neuron signals as $\Sigma_{\hat{S}}\in R^{364\times 364}$. Then, the solution $A\in R^{30\times 364}$ of the forward problem of each EEG segment could be obtained, which reflects the activity state of brain neurons. As shown in Figure 10b, we calculated the weight matrix set corresponded to the covariance matrices of all samples in two classes, respectively. Then, we calculated the average of weight matrix set of each class, and got the weight matrix A represents the fatigue and vigilant state, respectively. Thus, the *i*-th column of the matrix denotes the weight assignment of all channels by the *i*-th neuron during the conduction from the neurons to the scalp. In conclusion, in the different brain states, the weight assignment varies greatly, which can be seen in the third column of Figure 10b.

To explore the difference of the above weight assignment in fatigue and vigilant states intuitively, we utilized t-distribution [54] to fit the weight assignment and get a smoother distribution curve. The Kullback–Leibler Divergence (KL Divergence) defined as Equation (Equation 21) could quantify the difference between the two distributions *P* and *Q*, where *X* is the set of samples. In this study, the KL Divergence was used to quantify the difference of the weight assignment of all channels by the same neuron between different brain states; therefore, the influence of EEG covariance on different brain states can be observed intuitively. We calculated the KL Divergences of all neurons and the values fall in the range between 96.91 and 176.92. Then, these KL Divergence were normalized to the values between 0 and 1 for further comparison.
(21)$$D\left(P||Q\right)=\sum_{i\in X}P\left(i\right)\cdot \left[\log(\frac{P(i)}{Q(i)})\right]$$

As shown in Figure 11, we can obviously obtain the difference of weight assignment between fatigue and vigilant brain states. Considering the large number of neurons, we only show the activity difference of nine neurons. The blue curve represents the t-distribution diagram fitted by the weight assignment in the fatigue state, and the yellow curve denotes that in the vigilant state. According to the concept of t-distribution, the shape of the curve represents the concentration degree of weight assignment of all channels by each neuron we assumed. Moreover, the curves with different colors obtained from the *i*-th neuron represent the difference between the concentration degree of assignment with different brain states. As shown in Source_136 of Figure 11, we can see that the assignment of all channels by the 136th neuron is more concentrated in the fatigue state than the vigilant state, which has a significant difference. In Source_88, the assignment is more concentrated in the vigilant state, which is opposite to Source_136. In conclusion, there is an obvious difference between the weight assignment of two states, which can be seen in the t-distribution diagrams and KL divergence on the top of each diagram. Therefore, it is proven that the covariance matrices obtained from the different level of fatigue state greatly reflect the information of brain activity in different states, which provide the effective features for classification.

## 4. Conclusions

In this study, we proposed a novel spatial-temporal fusion method on the Riemannian manifold by calculating the covariance matrices of EEG signals, which can well support the driver fatigue detection with high effectiveness. Compared with other traditional methods considering the spatial covariance matrices, we utilized the temporal features of the covariance matrix obtained from EEG signals. In addition, we took advantage of two aspects of spatial characteristics: numerical characteristics after mapping and distance information, which outperform other methods, and the combination of these two aspects are proven to be complementary, which is better than only use one of them. Furthermore, this paper offers a new perspective of research on EEG signals modeling and further contributes to the extension of signal processing and modeling conception. As we mentioned before, the proposed EEG-based model has the potential to be used in practice, and we believe the new perspective could help to construct more intelligent or emotion-aware applications, which will contribute to the improvement of the quality of human life.

## Figures and Tables

**Figure 1 entropy-23-01298-f001:**
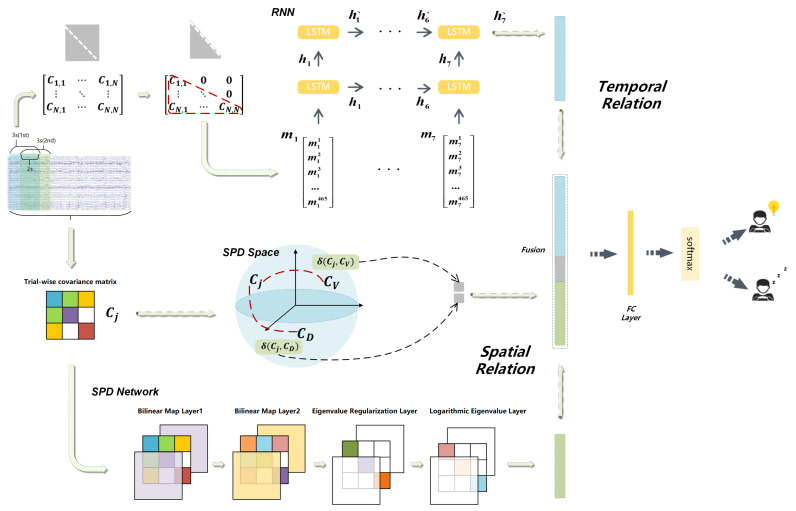
The framework of our model: In the SR domain, we fused the output of the SPD network and the distance information based on the Stein divergence as SR-domain features. In the TR domain: we fed the covariance matrices into a two-layer RNN with seven LSTM cells per layer, and define the output as TR-domain features. Then, the concatenated SR-domain and TR-domain features were transformed into linear features by the fully connected layer, and the features were used to give the final prediction by the softmax layer.

**Figure 2 entropy-23-01298-f002:**
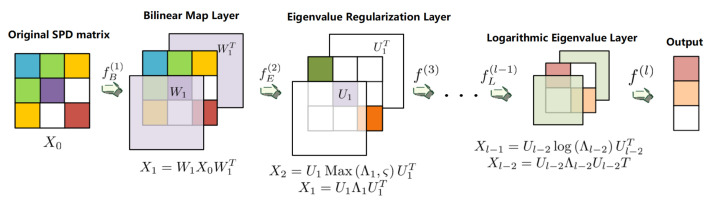
Architecture of the SPD network: The bilinear map layer consists of a transformation matrix *W* with full rank and its transpose. ζ of the second layer indicates a diagonal matrix, whose diagonal elements are the threshold value we set. In the logarithmic eigenvalue layer, we apply eigendecomposition to get the eigenvalues and eigenvectors of matrix $X_{i}$, which are $\Lambda_{i}$ and $U_{i}$.

**Figure 3 entropy-23-01298-f003:**
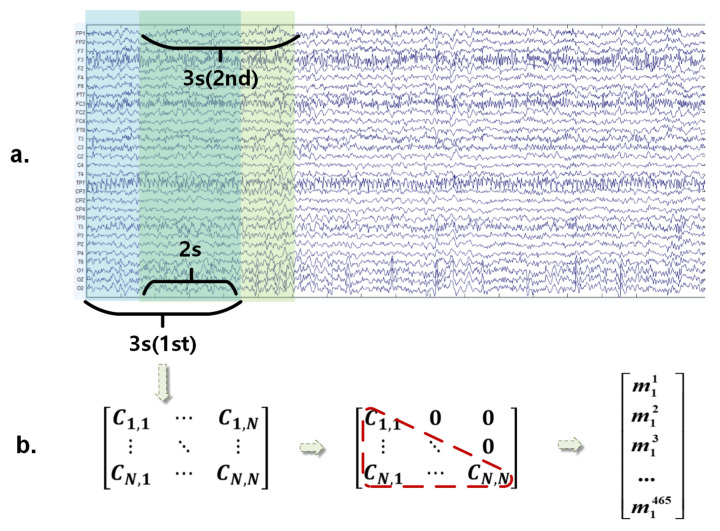
The scheme of EEG segmentation: (**a**) For time series of each trial, a 3 s sliding time window and a 1 s step are used to divide the EEG signals. (**b**) From the segmented result, the relations between different channels are captured by the covariance matrices we calculated, and we reshape them into vectors with one dimension, separately. The covariance value between the *i*-th channel and the *j*-th channel is indicated by the element C(i,j) of the covariance matrix.

**Figure 4 entropy-23-01298-f004:**
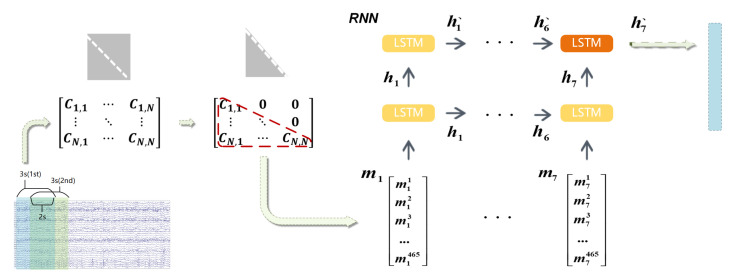
The features of the TR domain are processed by a 2-layer RNN with 7 LSTM units per layer, where $m_{i}$ means the flattened matrix of the *i*-th segment signals, and $h_{i}$ indicates the *i*-th hidden state. The $h_{7}^{{}^{\prime }}$ denotes the output of the RNN.

**Figure 5 entropy-23-01298-f005:**
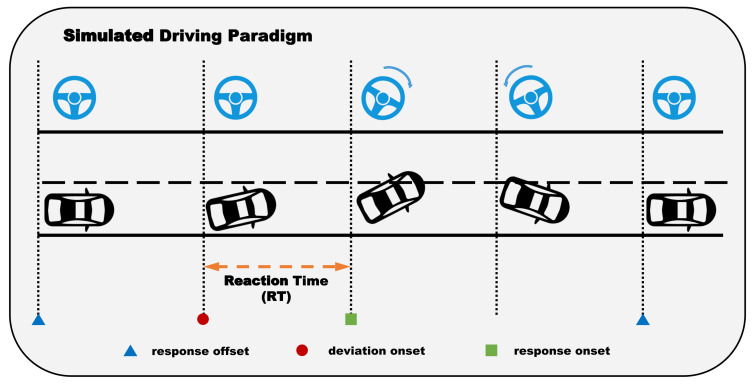
Simulated driving paradigm: Each lane-departure event is considered as the period from the first response offset to the second response offset, which contains the deviation onset and response onset of the current event. The time between the deviation onset and the response onset is defined as reaction time(RT), which is the main focus of our study. EEG signals were recorded during the whole task.

**Figure 6 entropy-23-01298-f006:**
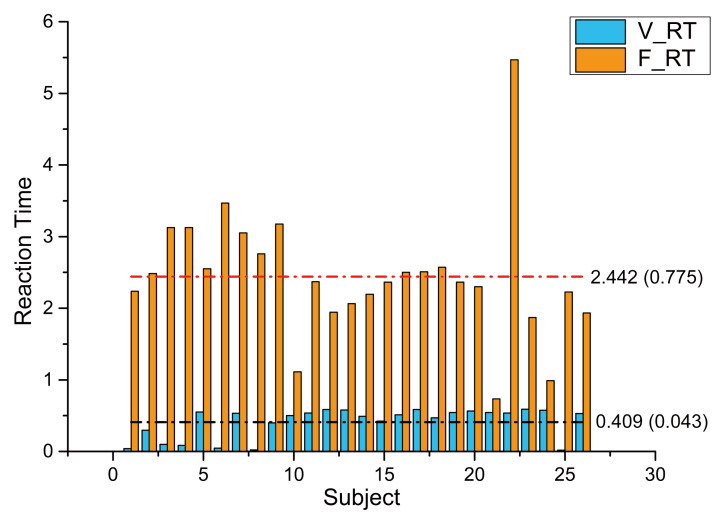
Comparison of the average RTs of 27 subjects in different driving states. The histogram with blue color (V_RT) represents the reaction time of the vigilant state, while the orange one (F_RT) represents the reaction time of the fatigue state. The horizontal dotted lines indicate the average RT of all subjects in different states. Besides, we marked the variance of reaction time, which are 0.775 and 0.043 of fatigue state and vigilant state, respectively.The average reaction time distribution of 27 subjects in the vigilant and drowsy driving state. The green histogram (V_Res) represents the response time of vigilant driving, while the purple histogram (F_Res) represents the response time of drowsy driving. The red and black horizontal dotted lines represent the average response time across all subjects in drowsy and vigilant driving, respectively. Besides, the variance of response time in drowsy and vigilant driving is 0.775 and 0.043.

**Figure 7 entropy-23-01298-f007:**
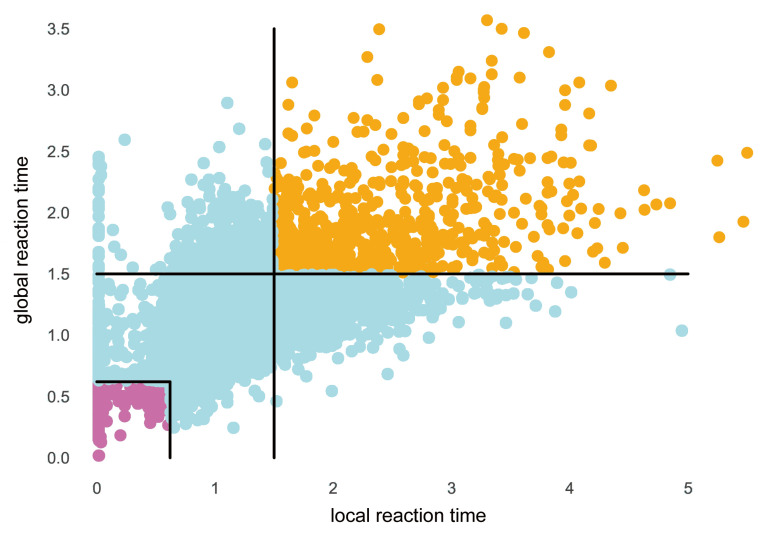
Scatter diagram of RT values of all trials: The horizontal axis represents the value of local reaction time, and the vertical axis represents the value of global reaction time; 739 trials were labeled as vigilant states, which were marked as purple points in the lower left corner; 694 orange points in the upper right corner indicated trials with fatigue state.

**Figure 8 entropy-23-01298-f008:**
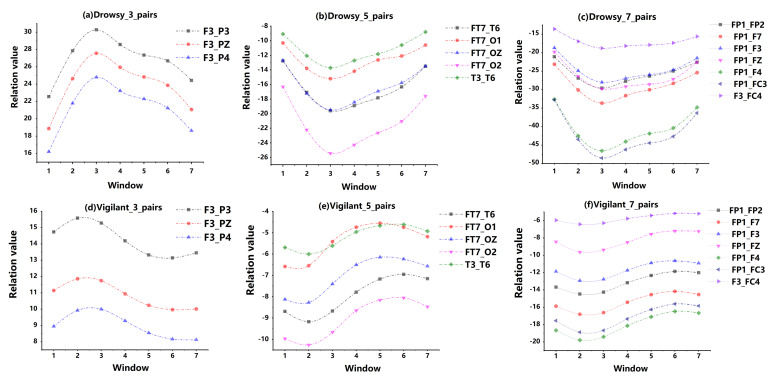
Trajectories of variations in the inter-channel relations over time in the situation of 3, 5, 7 channel pairs. In each graph (**a**–**f**), the horizontal axis represents the 7 successive time windows, and the vertical axis indicates the relation values between different EEG channels, which are the covariance of two channels. As seen, the curve shapes within the same class are very similar to each other, whereas significant dissimilarities exist between the different classes. For example, graphs (**a**,**d**) show the dynamic variations of the same combination of 3 EEG channel pairs in $M_{V}$ and $M_{D}$, respectively.

**Figure 9 entropy-23-01298-f009:**
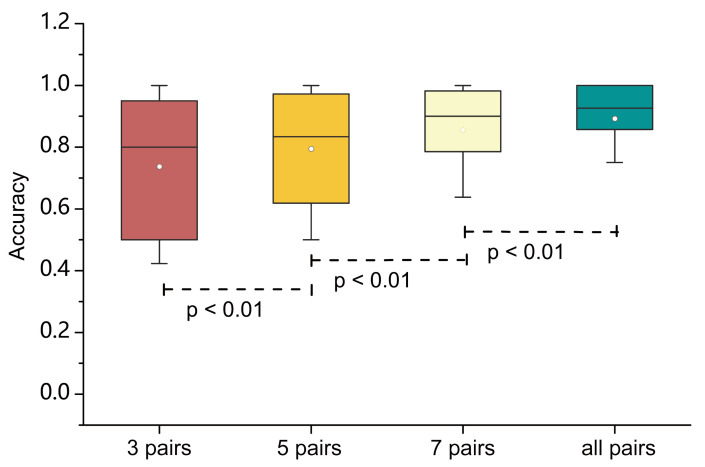
The box plots of classification accuracy of different numbers of channel pairs, which are 3, 5, 7 and all pairs. The *p*-values of the significance test are marked below the box plots.

**Figure 10 entropy-23-01298-f010:**
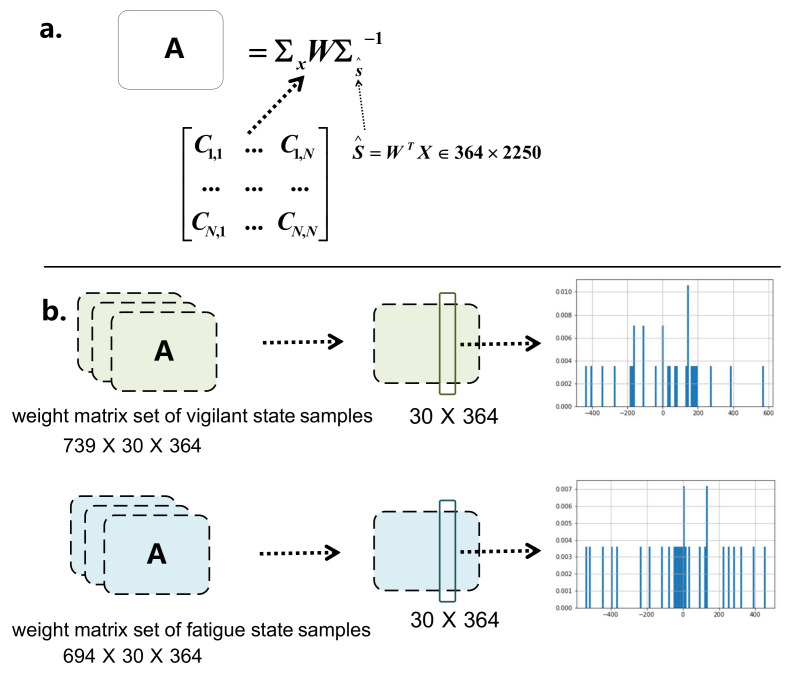
Diagram of brain state analysis based on covariance: (**a**) We assumed that the electrical activity is the matrix with the dimensions of 364×2250. (**b**) We calculated the weight matrix set A of fatigue and vigilant states, respectively. The histograms of the third column represent the weight assignment of all channels by the same neuron in two brain states.

**Figure 11 entropy-23-01298-f011:**
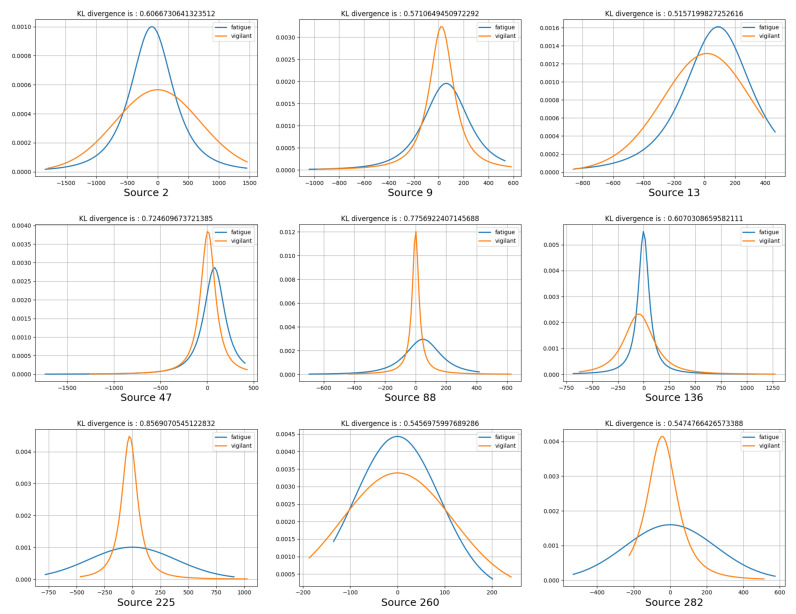
Fitted t-distribution diagrams of the weight assignment of all channels by the brain neurons, in which the blue curve denotes the fatigue state and the yellow curve denotes the vigilant state. The normalized KL divergences between two distributions are on the top of each diagram.

**Table 1 entropy-23-01298-t001:** Defined measurement of selected combinations of EEG channel pairs.

	Γ	Θ	Φ	ζ
2_pairs_combination_1	1.0298	1.0521	0.0302	68.9370
2_pairs_combination_2	1.0603	1.0844	0.0773	29.7875
2_pairs_combination_3	1.0514	1.0100	0.1595	12.9241
3_pairs_combination_1	1.0959	1.0960	0.0238	92.0631
3_pairs_combination_2	1.0665	1.1007	0.0523	32.9318
3_pairs_combination_3	1.0620	1.0859	0.1023	20.8332
5_pairs_combination_1	1.0500	1.0762	0.0591	34.7570
5_pairs_combination_2	1.0545	1.0847	0.0651	31.8253
5_pairs_combination_3	1.0875	1.0686	0.0907	23.5354
7_pairs_combination_1	1.0651	1.0704	0.0674	36.1222
7_pairs_combination_2	1.0722	1.0806	0.0596	31.3334
7_pairs_combination_3	1.0651	1.0834	0.0931	22.7714

**Table 2 entropy-23-01298-t002:** Comparison of our method with other methods in terms of various evaluation criteria (accuracy, specificity, sensitivity, and F1 score).

		Accuracy (%)	Specificity (%)	Sensitivity (%)	F1 (%)
COV_CNN	Mean	81.877 **	74.149 **	72.435 **	77.204 **
	Variance	3.189	12.418	10.775	6.001
CNN_RNN	Mean	79.103 **	73.036 **	72.487 **	74.749 **
	Variance	5.028	13.059	10.668	7.476
FRE	Mean	75.155 **	70.708 **	74.218 **	73.047 **
	Variance	3.894	12.878	10.640	6.882
TRDC	Mean	86.238 **	74.577 **	80.719 **	80.966 *
	Variance	2.377	11.987	10.375	7.741
SDTR	Mean	89.280 *	78.177	82.327 *	85.368
	Variance	1.711	10.891	8.568	5.077
SNTR	Mean	91.042	72.571	90.800	81.321
	Variance	1.369	14.013	4.629	7.118
OurMethod	Mean	93.834	83.168	85.863	85.686
	Variance	0.902	12.398	8.948	7.847

(*p*:obtained with paired samples *t*-test; *: p≤0.05, **: p≤0.01).

## Data Availability

Not applicable.

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
