# Peer review of "Fatigue Detection with Spatial-Temporal Fusion Method on Covariance Manifolds of Electroencephalography"

_entropy, 2021, doi:10.3390/e23101298_

Round 1
Reviewer 1 Report
Authors presented a study in which they detected fatigue from high density EEG signals, by using spatial-temporal fusion methods. The work is well presented, and very interesting. Authors should address just some concerns before the publication:
- Introduction: Authors should cite other works rather than only covariance based ones. E.g. https://www.mdpi.com/1424-8220/21/7/2369, or similar.
- Did the authors asked for an ethical approval of the study?
- Authors should discuss the implications for an online use of their methodology, and potentially with less channels.
- How the authors can be sure that they are not classifying any confound effect, rathen than fatigue? e.g. sensory motor rythms, or other mental states? Please discuss this aspect.
Author Response
Dear reviewer,
Thank you for the time and effort you have dedicated to providing insightful feedback on ways to strengthen our paper.
Please see the attachment about the point-by-point response.

Reviewer 2 Report
The paper develops a new method aimed at acknowledging fatigue sates from EEG data in health subjects. The study is interesting and the research has some scientific soundness. I have some specific and general concerns that are listed below.
Specific comments
The main table of results, namely Table 2, presents the comparison of the results obtained by the proposed method and other methods present in the literature. For any value of accuracy, specificity and sensitivity the p-value is reported by asterisks in alla cases, except for the values of the new and SNTR methods. Is that meaning that the results are not enough meaningful statistically in these cases? Please explain it better.
Figure 11 reports the comparison of the T-distribution diagrams of vigilant (orange) and fatigue (blue) states. In some cases the orange curves show larger (sources 8, 22, 53 and 81) peaks than the blue ones and in other the opposite occurs (sources 1, 12, 13, 16 and 24). Does it have any meaning? Moreover, the KL divergence is reported in the top of any panel. The values fall in the range 132-160 approximately. These values can be large or small depending on the scale of variation of the parameter. Can the author perform some kind of normalization in order the values can be fruitfully compared among each other?
General comments
The authors claim that their method outperforms pre-existing ones, and this seems correct on the basis of the results reported in the paper. Anyway, nothing is said about how the new method can be used in practice. For example, has the computing time of the algorithm been evaluated? Is the method feasible for real applications, for example may it be implemented in safety systems of cars? Moreover, the experiment has been conducted for a sample of 27 subjects, is this enough to conclude that it is applicable? I suggest to develop in future studies further testing of the algorithm on other databases of reference in the field.
A deep reading by a mother tongue lecturer can improve the quality of the paper.
The paper can be accepted after solving the above unclear points.
Author Response

(The authors gave the same response as above.)
